# S-phase transcriptional buffering quantified on two different promoters

Sharon Yunger[1,3], Pinhas Kafri[1,3], Liat Rosenfeld[2,3], Eliraz Greenberg[1,3], Noa Kinor[1,3], Yuval Garini[2,3], Yaron Shav-Tal[1,3]

Imaging of transcription by quantitative fluorescence-based techniques allows the examination of gene expression kinetics in single cells. Using a cell system for the in vivo visualization of mammalian mRNA transcriptional kinetics at single-gene resolution during the cell cycle, we previously demonstrated a reduction in transcription levels after replication. This phenomenon has been described as a homeostasis mechanism that buffers mRNA transcription levels with respect to the cell cycle stage and the number of transcribing alleles. Here, we examined how transcriptional buffering enforced during S phase affects two different promoters, the cytomegalovirus promoter versus the cyclin D1 promoter, that drive the same gene body. We found that global modulation of histone modifications could completely revert the transcription down-regulation imposed during replication. Furthermore, measuring these levels of transcriptional activity in fixed and living cells showed that the transcriptional potential of the genes was significantly higher than actual transcription levels, suggesting that promoters might normally be limited from reaching their full transcriptional potential.

## Introduction

Transcription is a key event in the gene expression pathway. Imaging of transcription in living cells by the use of fluorescence techniques has become an important tool in our understanding of the dynamic expression of genes, and has been providing unique information, in parallel to data obtained from biochemical, molecular, and bioinformatics approaches (Tutucci et al, 2018). Transcription kinetics can be measured in living mammalian cells on the single-gene and single mRNA levels (Chubb et al, 2006; Yunger et al, 2010; Lionnet et al, 2011; Martin et al, 2013; Coulon et al, 2014; Park et al, 2014; Senecal et al, 2014; Kalo et al, 2015; Kafri et al, 2016).

An important question in the field relates to how cells control mRNA transcription levels throughout the cell cycle. We have previously followed transcription from single alleles during the different phases of the cell cycle. We used a cell system that allowed real-time tagging of mRNAs transcribed from a single *cyclin D1* (*CCND1*) gene in living human cells (Yunger et al., 2010, 2013; Kafri et al, 2016). Using isogenic single-gene genomic integrations, we examined *CCND1* transcription under the control of two promoters, the endogenous *CCND1* promoter and the cytomegalovirus (CMV) promoter. We found that the levels of active mRNA transcription were significantly modulated after DNA replication (S phase). Transcription that occurred after replication was easily visualized in this system since the duplicated transcribing genes on the sister chromatids were detected as gene doublets. This analysis revealed a drastic reduction in the transcription levels of these two alleles from after replication up until cell division. Specifically, the transcriptional output of the two *CCND1* alleles after replication was 50% lower than that in the one allele in G1 before replication. Together, the output of the two alleles was similar to the mRNA production of one allele before replication, such that CCND1 mRNA levels remained relatively constant during the cell cycle.

Does mRNA expression change during the cell cycle? The general notion from yeast and mammalian cells has been that cells can buffer the change in gene dosage brought about during replication and accordingly regulate and balance mRNA and protein expression levels (Elliott & McLaughlin, 1978; Barnes et al, 1979; Skog & Tribukait, 1985). A more recent study in which mRNA levels were quantified in single cells during the cell cycle has also shown for several genes that there is a 50% drop in the number of actively transcribing alleles after replication (Padovan-Merhar et al, 2015). This study examined cell volume and its influence on transcription and concluded that there must be a mechanism for reducing transcription after replication to maintain constant transcription throughout the cell cycle. This suggested mechanism was not dependent on the volume-compensating mechanism, which they discovered. In another study that examined the dosage compensation effect on two mouse genes throughout the cell cycle, it was found that the *Oct4* and *Nanog* genes show a decrease in the transcriptional activity of each allele following replication (Skinner et al, 2016). Mathematical modeling showed that the rates of gene activation were important in determining the buffering effect. Importantly, a biochemical mechanism responsible for the buffering

[1]The Mina and Everard Goodman Faculty of Life Sciences, Bar Ilan University, Ramat Gan, Israel [2]Department of Physics, Bar Ilan University, Ramat Gan, Israel [3]Institute of Nanotechnology, Bar Ilan University, Ramat Gan, Israel

Correspondence: Yaron.Shav-Tal@biu.ac.il

effect was recently revealed in a global study performed in yeast cells (Voichek et al, 2016a) and showed that gene expression homeostasis is maintained over the cell cycle, such that an increase in gene dosage does not change the expression levels of many genes. The molecular process behind this buffering mechanism was attributed to the acetyltransferase Rtt109 and its chaperone Asf1, which acetylate histone H3 on K56 residue and K9 residues. Specific examination of these modifications in mutant cells showed that H2K56 is the modification that is crucial for the buffering effect.

In this study, we further quantified the effect of the buffering of mRNA transcription due to the passage of the cell through replication, in mammalian cells, by tempering with the buffering process. We used the human *CCND1* single-gene cell system that we previously generated, in which the same gene is expressed from the same genomic locus but under the control of two different promoters—the CMV promoter versus the cyclin D1 promoter. We found that the buffering effect and the 50% reduction in CCND1 transcription could be completely eliminated when interfering with histone deacetylation. Namely, we could cause the duplicated genes after replication to transcribe to the same levels just as the gene during G1 phase. Moreover, under these non-buffering conditions, the genes were transcribing at higher levels than usual, with a prominent effect on the CMV promoter-driven transcription compared with a limited effect on the CCND1 promoter. Our findings also reveal that transitioning between low and high transcription levels due to histone modifications can be extremely rapid. Altogether, this study suggests that the full transcriptional potential of a promoter can be limited by the cell throughout the cell cycle.

# Results

### Experimental system for detecting active genes before and after replication

We previously generated a cell system in which a *CCND1* gene was integrated as a single copy allele into human HEK293 cells using the Flp-In recombination system (Yunger et al, 2010). Transcription kinetics on this gene were visualized and quantified using RNA FISH and live-cell imaging. Labeling of the CCND1 mRNA was obtained using a series of MS2 sequence repeats inserted into the long 3′-UTR of *CCND1* (Fig 1A). By co-expressing a GFP-fusion MS2 coat protein (MS2-CP-GFP), we achieved fluorescent tagging of the mRNAs produced from this gene in living cells and could follow single-gene transcription in real time (Fig 1B). Using RNA FISH with a fluorescent probe that hybridized to the MS2 repeats, we could quantify the transcriptional output of the *CCND1-MS2* genes in fixed cells (Fig 1C).

Two unique cell clones were generated containing an integrated *CCND1* gene under the control of either the endogenous *CCND1* promoter (gene termed: $D1^{CCND1pr}$) or under the control of the CMV promoter ($D1^{CMVpr}$) (Yunger et al, 2010). The number of nascent transcripts transcribed on the $D1^{CMVpr}$ gene was roughly twice that on the $D1^{CCND1pr}$ gene (14 ± 4 versus 7 ± 4 nascent transcripts/gene). This implied that several RNA polymerase II (Pol II) enzymes were engaged in active transcription along the genes, and modeling the data showed that successful Pol II recruitment to the promoter was

twice as high when driven by the CMV promoter (Yunger et al, 2010). Importantly, we could follow transcriptional activity on these genes throughout the cell cycle. It was possible to specifically detect cells after replication of the *CCND1-MS2* alleles, since they contained adjacent duplicated and active *CCND1-MS2* genes on the sister chromatids (Fig 1D). Surprisingly, the number of nascent transcripts being transcribed by the duplicated genes was 50% lower than that in cells in G1, meaning that the potency of the gene to transcribe after replication had been significantly disrupted by passage through replication. This was the case for both promoters, suggesting that a similar mechanism is regulating and restricting transcriptional potential after replication on the way to cell division. Later studies observed this buffering phenomenon for other genes and in other systems (Padovan-Merhar et al, 2015; Voichek et al, 2016a).

### Enhanced transcription of the $CCND1^{CMVpr}$ gene following HDAC inhibition

The transcriptional down-regulation observed after replication for both promoters indicates that a global mechanism is at play rather than sequence-specific influences on transcription. The homeostasis mechanism studied in yeast showed that this is regulated through histone acetylation (Voichek et al, 2016a). We examined how inhibiting the deacetylation of histones, thereby increasing acetylation levels, (i) influences the levels of transcription from the two genes, and (ii) what are the levels of transcriptional activation that can be reached by the two promoters during the cell cycle. Cells were treated with the HDAC inhibitor trichostatin A (TSA) (Finnin et al, 1999). Since the two genes were integrated at the same gene locus, we could compare the response of the two promoters with similar changes in the chromatin environment. First, we calibrated the cell system to conditions that do not affect cell viability, as high TSA levels can lead to apoptosis (Toth et al, 2004). We proceeded with a treatment of 100 nM for 7 h. In addition, since TSA can cause cell cycle blockage, we tested the cell cycle profile by FACS and found no significant change in the percentage of cells in G1 and a minor increase in the percentage of G2 cells on account of cells in the S phase (Fig S1A). The percentage of cells with single or duplicated active *CCND1-MS2* genes changed slightly after TSA treatment (Fig S1B).

We performed RNA FISH on the *CCND1-MS2* mRNAs transcribed from the CMV promoter ($D1^{CMVpr}$ gene) under TSA treatment and found enhanced fluorescence signal of the active transcription sites (single genes) for cells in G1 (Fig S2A). Importantly, the fluorescence intensity on the duplicated genes (after replication of the alleles) (Fig 2A), which was normally faint in untreated cells because of the reduction in transcriptional activity, was much stronger after TSA treatment (Fig 2A). A second observation was the physical separation between the sister chromatids detected in 25% of the cells with duplicated genes (Figs 2A and S1C). RNA-FISH quantification showed a twofold increase of nascent mRNAs for cells in G1 (one allele), showing 15 ± 5 nascent transcripts per active gene in untreated cells compared with 27 ± 11 nascent transcripts per active gene in cells treated with 100 nM TSA (Fig 2B). The enhanced activity of the gene was accompanied with an increase in the total number of *CCND1-MS2* mRNAs in the cells (Figs 2C and S2B).

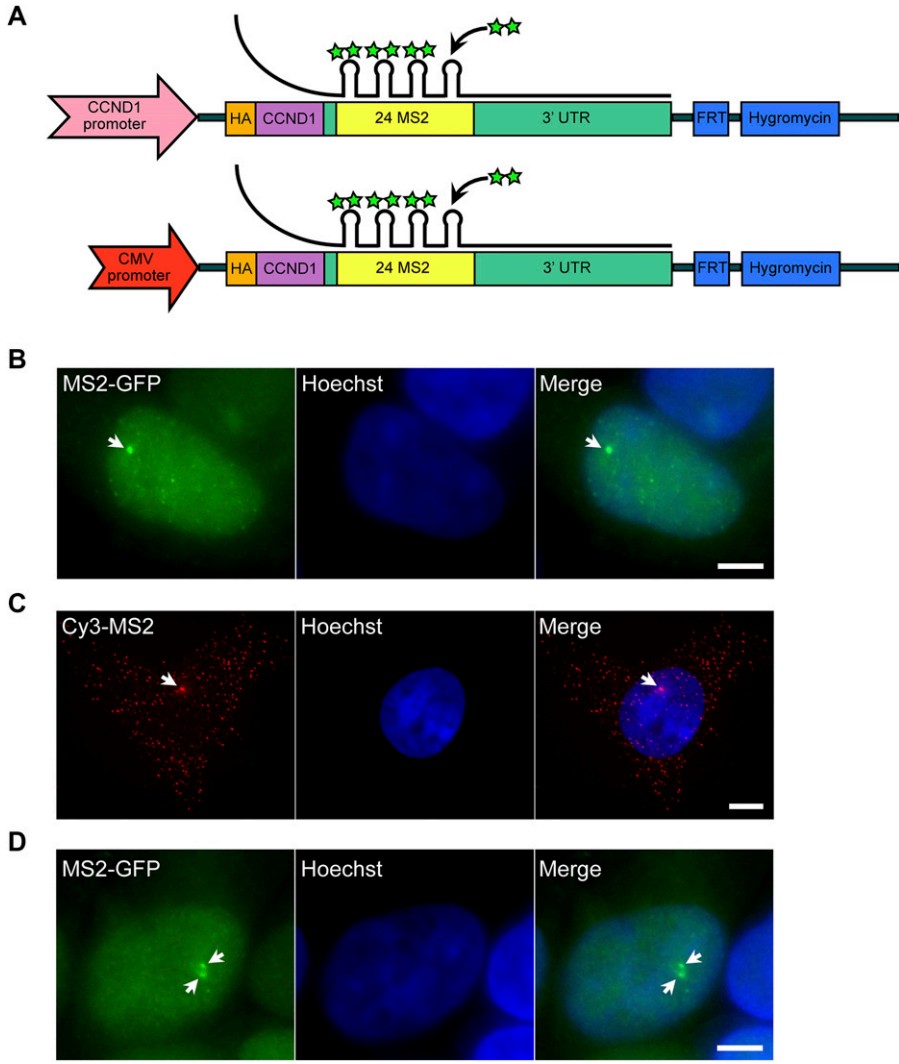

**Figure 1. Cell system for following and quantifying single-gene transcription in fixed and living cells.**
**(A)** The *cyclin D1* genes used in the study were stably transfected as single-copy genes into HEK293 cells using the Flp-In approach (Yunger et al., 2010, 2013). One gene is under the control of the *CCND1* promoter and the other version is regulated by the CMV promoter. The coding region contains an HA tag and is followed by 24× MS2 sequence repeats and the 3′-UTR of *CCND1*. The MS2 repeats form secondary structures that can be bound by MS2-CP-GFP (green stars) for tracking transcription in living cells. **(B)** A transcribing *CCND1-MS2* gene under the control of the CMV promoter tagged with MS2-CP-GFP. Arrow points to the site of transcription and bright green dots are single CCND1 messenger RNPs (mRNPs). Hoechst DNA stain is in blue. **(C)** RNA FISH with a Cy3-labeled probe that hybridizes with the MS2 region of the mRNA. Arrow points to the site of transcription and red dots are single CCND1 mRNAs. **(D)** An MS2-CP-GFP–expressing cell showing two adjacent *CCND1* alleles (arrows). Scale bar, 5 μm.

## Postreplication transcriptional down-regulation is relieved by HDAC inhibition

We then examined the transcriptional activity of the genes in cells that had passed through replication, which were identified by the presence of the duplicated $D1^{CMVpr}$ alleles. Here too, RNA FISH quantification showed an increase in nascent and total *CCND1-MS2* mRNA in the cells in response to TSA (Fig 2B and C). Interestingly, in TSA-treated cells, both the duplicated genes after replication of the alleles and the single gene before replication, transcribed at the same high levels, in contrast to the duplicated genes in untreated cells that normally had reduced transcriptional activity. Namely, after TSA treatment, nascent mRNA numbers on the duplicated genes increased fivefold from 6 ± 3 transcripts/gene in untreated cells to 29 ± 13. We found this increase to be dose dependent, increasing to 22 ± 11 nascent mRNAs for 30 nM TSA (Fig S2C). Altogether, there was a fivefold induction of activity for the duplicated genes after TSA compared with a twofold induction for single genes (G1), showing the negative effect of TSA on the buffering process (Fig S2D).

## The cyclin D1 promoter-driven gene responds to the inhibition of histone deacetylase (HDAC) activity

We next examined the effect of TSA on the same *CCND1-MS2* gene but under the control of the endogenous *CCND1* promoter. There was a moderate increase in nascent mRNAs (Fig 3A and B) and total mRNA (Figs 3C and S2B) after TSA compared with the same gene transcribing from the same genomic locus under the control of the CMV promoter (Fig 2). For single genes in G1, there was an increase from 8 ± 2 to 10 ± 3 nascent mRNAs after TSA and from 4 ± 3 to 9 ± 4 for the duplicated genes. Here too, the relatively higher levels of transcription of the single and duplicated alleles after TSA were similar, showing that the promoter had acquired a higher potential to initiate transcription events because of the activity of the HDAC inhibitor. We note that although there is a decrease in the transcriptional output of the alleles after replication from 8 ± 2 nascent mRNAs for the single alleles in G1 to 4 ± 3 nascent mRNAs for doublets after replication (before TSA treatment), the cellular levels of the mRNAs remain generally the same, since each of the two alleles is simultaneously

**A**

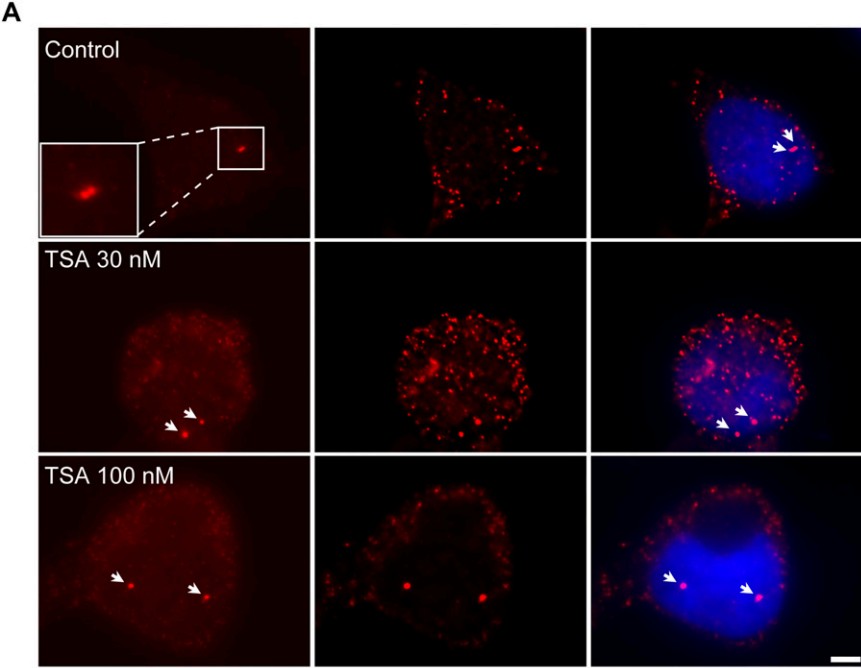

**Figure 2. TSA induces transcription from CCND1-MS2 alleles under CMV promoter control.**
**(A)** RNA FISH showing the intensity of duplicated CCND1-MS2 sites of transcription (arrows) in control and TSA-treated cells (7 h). The middle column shows the deconvolved image of this cell. Total mRNAs counted: top, 136; middle, 116; bottom, 196. Scale bar, 5 μm. **(B)** Single molecule RNA FISH quantification of the number of nascent CCND1-MS2 mRNAs on the active sites of transcription (single and doublets) in control and TSA-treated cells (7 h). Number of alleles quantified (n) is marked in red below each box. **(C)** Single molecule RNA FISH quantification of the total number of CCND1-MS2 mRNAs in the same in control and TSA-treated cells. In the boxplots, the median is indicated by a line in the box, the box represents the interquartile range, the whiskers represent the maximum and minimum values, and dots represent outliers. Number of cells quantified (n) is marked in red below each box.

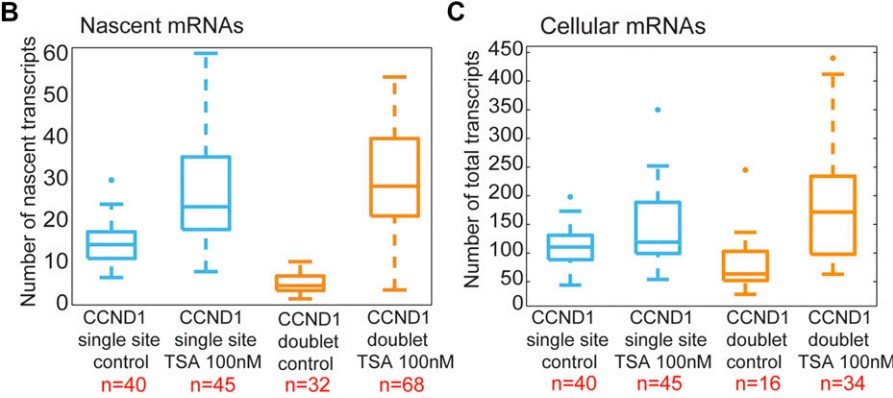

producing half of the mRNAs that a single allele does. Altogether, these data show that TSA can counteract the down-regulation of transcription occurring because of the passage of the cell through replication and S phase and thereby activate transcription to levels that are even higher than the cells in G1 (summarized in Fig S3).

### Premature separation of transcribing sister genes does not attenuate high transcription levels

As presented in the RNA FISH images Fig 2A, the treatment with TSA caused the duplicated genes, normally adjacently situated on sister chromatids, to be found far from each other in the nuclear volume in 25% of the cells with duplicated alleles (Figs 4A and S1C). This phenomenon could also be seen by MS2-CP-GFP labeling (Fig 4B). The fact that hyperacetylation can cause premature sister chromatid separation has been documented at the chromosome level (Magnaghi-Jaulin et al, 2007). We found that the distances correlated

with TSA concentration (Fig 4C). Measuring the separation distances in TSA-treated cells (Fig 4D) showed that the average distance between the physically separated genes was 6 μm compared with 1 μm in cases where the genes did not separate.

Analyzing the differences between the transcription levels of the two duplicated alleles within the same cell showed high variation between the two active sites in TSA-treated cells (mean difference: 15 ± 11 transcripts) compared with much lower variation in untreated cells (mean difference: 3 ± 3 transcripts), (Fig S4). When we examined whether the physical separation between the genes per se might be driving these differences in transcription activity, we did not find a correlation between the distance between the duplicated genes and the levels of variation in transcription of the two alleles (Fig S5). Also, adjacent duplicated genes under TSA treatment could show large variations between the two alleles (green dots in the left-hand part of the plot in Fig S5), meaning that the distance was not the factor determining the transcriptional activity. Finally, although differences

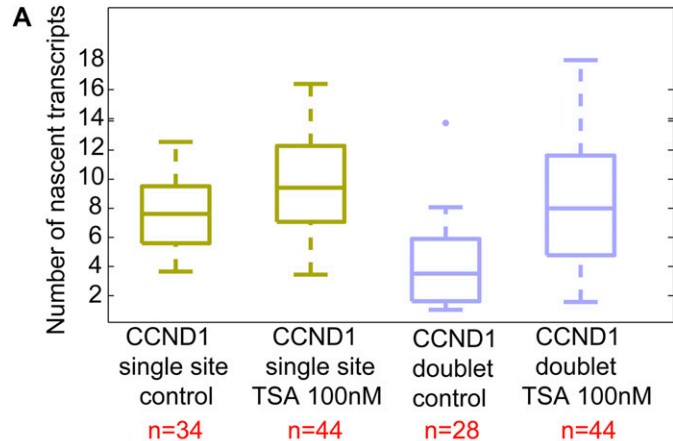

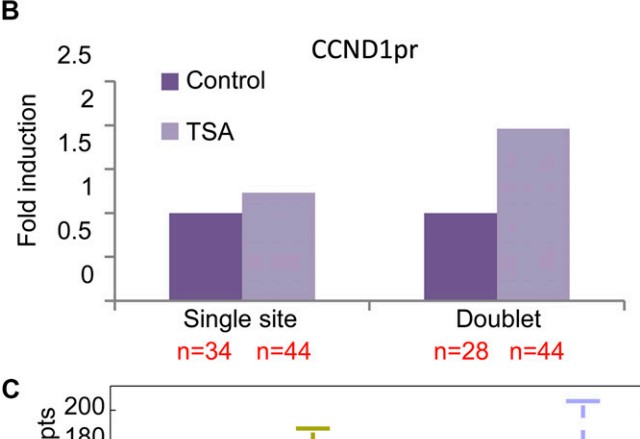

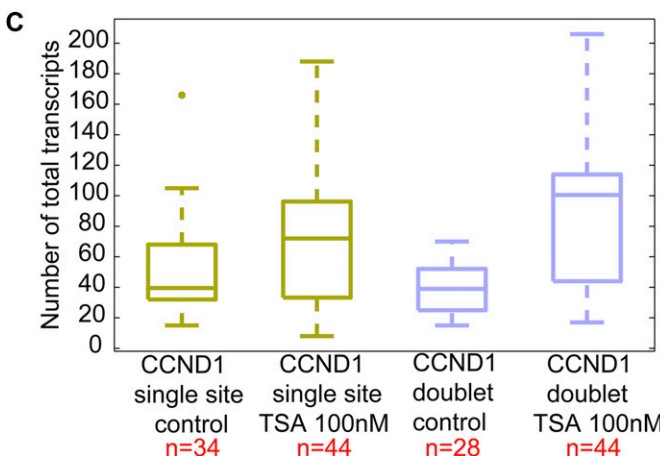

**Figure 3. TSA induces transcription from CCND1-MS2 alleles under CCND1 promoter control.**
**(A)** Single molecule RNA FISH quantification of the number of nascent CCND1-MS2 mRNAs on the active sites of transcription (single and doublets) in control and TSA-treated cells (7 h). **(B)** Fold-induction levels of transcriptional activity of single genes and duplicated gene after TSA treatment compared with untreated cells (designated as 1). $P$ single = 0.009 and $P$ duplicated = 1.87549 × 10$^{-5}$. **(C)** Single molecule RNA FISH quantification of the total number of CCND1-MS2 mRNAs in the same in control and TSA-treated cells. Number of alleles quantified (n) is marked in red below each box.

in gene activity can be attributed to the nuclear positioning of genes in the nuclear space (peripheral or central location in the nucleus), no correlation was observed (Fig S6). Altogether, we concluded that there was no correlation between the physical separation of the sister chromatids and their levels of transcriptional activity.

Previously, we showed that the replication of the locus harboring the *CCND1-MS2* genes occurred during the mid/late S phase (Yunger et al, 2010). To examine whether the separation between the duplicated genes also took place during S phase, we labeled cells with 5-ethynyl-2′-deoxyuridine (EdU), a thymidine analogue that incorporates into replicating DNA during S phase. Pulsing the cells with a fluorescent form of EdU that enabled the direct detection of replicating cells in a population of TSA-treated cells, we could detect cells in S phase, but they only had single genes or duplicated genes that were in close contact (Fig S7A). Cells that had separated genes did not stain with EdU, meaning that the separation did not occur immediately after replication but probably only after replication had ended. Indeed, only cells stained with the G2 marker, CENP-F, showed separated sites (Fig S7B). This meant that the separation was a rather late event in the cell cycle, and that even so, under TSA conditions, these genes continued to transcribe at very high levels all the way to cell division.

To examine whether the mere separation of the duplicated genes could affect the levels of transcription, the analyzed cells with duplicated genes were divided into two subpopulations based on distances. Cells in which there were large distances between the genes had 33 ± 7 nascent transcripts associated with the transcribing genes, whereas cells with adjacent genes had 24 ± 9 nascent transcripts (Fig S8). When we plotted the number of nascent transcripts versus the distance, we found that in untreated cells, the transcription levels were low when the genes were adjacent (Fig S9). However, when the TSA-treated cells were examined, we found two subpopulations. For adjacent genes (up to 2 $\mu$m), the complete range of transcription levels was observed, whereas for the separated genes, there were only elevated transcription levels (Fig S9). This meant that a cell in which the genes had undergone separation would always have high activity from these genes, since both effects are driven by TSA. But even in cells that had not undergone separation, the genes were capable of reaching high transcription levels, suggesting that attaining the highest levels of transcriptional activity did not depend on physical separation. This was also supported by the live-cell movies in which we could see high activation levels when duplicated genes were adjacent (see below).

**Live-cell imaging of gene activity reveals that changes in chromatin can rapidly relieve the reduction in transcription imposed during S phase**

To measure the kinetics of the changes in transcription levels due to the TSA treatment, we followed the transcriptional activation in living cells after the addition of TSA. A slow and gradual buildup of MS2-CP-GFP fluorescence on the single active gene (G1) was observed, which took several hours after TSA addition (Fig 5A and Video 1). We managed to image some cells with duplicated alleles and under TSA treatment. When the transition of the duplicated genes from low levels of transcription to high levels following TSA treatment was imaged, a sudden burst in activation was observed (Fig 5B and Video 2). Examining the time-lapse data of Video 2 shows that the burst occurred in a very short time frame between the 50 and 60 min time points, namely, moving from a relatively low status of activity to very high activity occurs in a few minutes. Moreover, Video 3 shows a single gene that had responded to TSA by transcribing at high levels, but when the gene transitioned to the duplicated state, the levels of

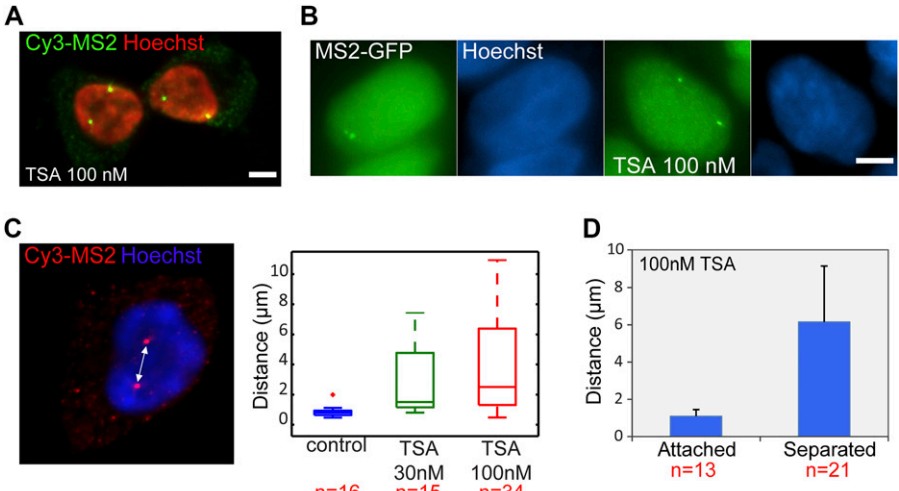

**Figure 4. TSA causes premature chromatid sister separation.**
**(A)** RNA FISH with a Cy3-labeled probe that hybridizes with the MS2 region of the mRNA (green), showing duplicated and separated CCND1-MS2 alleles on the background of Hoechst DNA staining (pseudocolored red) after TSA treatment (7 h). Scale bar, 5 μm. **(B)** Transcribing *CCND1-MS2* genes under the control of the CMV promoter tagged with MS2-CP-GFP showing adjacent (left) and separated (right) sister genes after TSA treatment (7 h). Hoechst DNA stain is in blue. **(C)** RNA FISH image showing separated genes and the distance between them. The boxplots show the measured distances under control and TSA treatments (7 h). **(D)** Average distances between duplicated alleles in TSA-treated cells (7 h). Number of cells quantified (n) is marked in red below the plots.

transcription were significantly reduced. A few minutes later, the duplicated alleles were observed transcribing at high levels comparable to the intensity of the single gene (Fig 5C). This demonstrates (i) that the decline in transcription during duplication still occurs but for a short and transient period and (ii) that the kinetics of transition between low and high transcription levels can take place very rapidly.

## Discussion

mRNA expression levels can change during the cell cycle and must account for the duplication of the genetic content in the cell from S phase through G2 until cell division. Several studies that used single-molecule mRNA FISH found a 50% reduction in transcription levels after replication for a number of mammalian genes (Yunger et al, 2010; Padovan-Merhar et al, 2015; Skinner et al, 2016). A more global study in yeast demonstrated that an mRNA transcription buffering mechanism is at play, which through acetylation/deacetylation events on histone H3 ensures that the increase in gene dosage is accounted for by a reduction in mRNA transcription levels, thus providing relatively constant levels for many types of mRNA during the cell cycle (Voichek et al, 2016a).

We have used a single-gene system that allows the detection of transcription in fixed and living cells, to follow mRNA transcription through the cell cycle (Yunger et al., 2010, 2013; Rosenfeld et al, 2015; Kafri et al, 2016). Previously (Yunger et al, 2010), we quantified the transcriptional output of the *CCND1* gene under the control of two very different promoters. Live-cell imaging showed that the *CCND1* gene under its endogenous promoter fluctuates between "ON" and "OFF" states, each lasting up to many minutes. This pulsatile behavior has been called "transcriptional bursting" and has turned out to be an important mode of transcription in eukaryotes to fine tune the level of transcription, mainly in the case of highly regulated genes, such as *CCND1*. On the other hand, the *CCND1* allele, driven by the CMV promoter, did not show any periods of gene inactivity, rather the gene was constantly in an "ON" state, except when undergoing cell division, as expected from an overexpression state.

Using quantitative RNA FISH, we found that there was at least a 50% reduction in the mRNAs transcribed from these genes after replication, for both of the promoters. This meant that the mRNA output of the duplicated alleles (after their replication) was comparable to the mRNA output of the single allele (before replication), thus demonstrating the buffering effect discussed above. In this study, we utilized protein acetylation/deacetylation to temper with the transcription driven from two promoters in mammalian cells during the cell cycle. We used TSA, which is an HDAC inhibitor that can cause histone hyperacetylation, chromatin decondensation, and gene activation. We found that preventing histone deacetylation led to an increase in the transcription levels of the duplicated alleles, thereby circumventing the buffering effect imposed during replication. In addition, the levels of transcription from the single gene were increased as well, when deacetylation was inhibited. In other words, for cells before replication, the genes exhibited expression levels that were significantly higher than normal untreated conditions. Furthermore, for cells in S/G2, the duplicated genes also showed high levels of transcription, which were similar to the transcription in the G1 cells, in stark contrast to the drastic reduction in transcription seen after replication under regular growth conditions.

The live-cell data showed that TSA treatment caused a gradual increase in transcriptional activity for the alleles before replication. In contrast, following the duplicated alleles showed genes that were barely transcribing (due to passage through replication), and then a dramatic burst in transcription merely several minutes after, suggesting that TSA treatment caused a rapid and global change in the chromatin landscape of the gene allowing it to reach extremely high levels of transcription, which are usually not observed. These live-cell data also give a feeling for the time scale of histone modification effects on transcriptional activation. These observations in live mammalian cells agree with the mechanism proposed for gene dosage homeostasis in yeast cells (Voichek et al, 2016a), namely, that Rtt109 leads to H3K56 acetylation, which gets incorporated into the DNA during replication, and at the end of S phase, this modification is removed by deacetylases (Voichek et al, 2016b). Our movies show that under TSA treatment, the duplicated genes can remain in the low expressing state for a very short time after replication and that

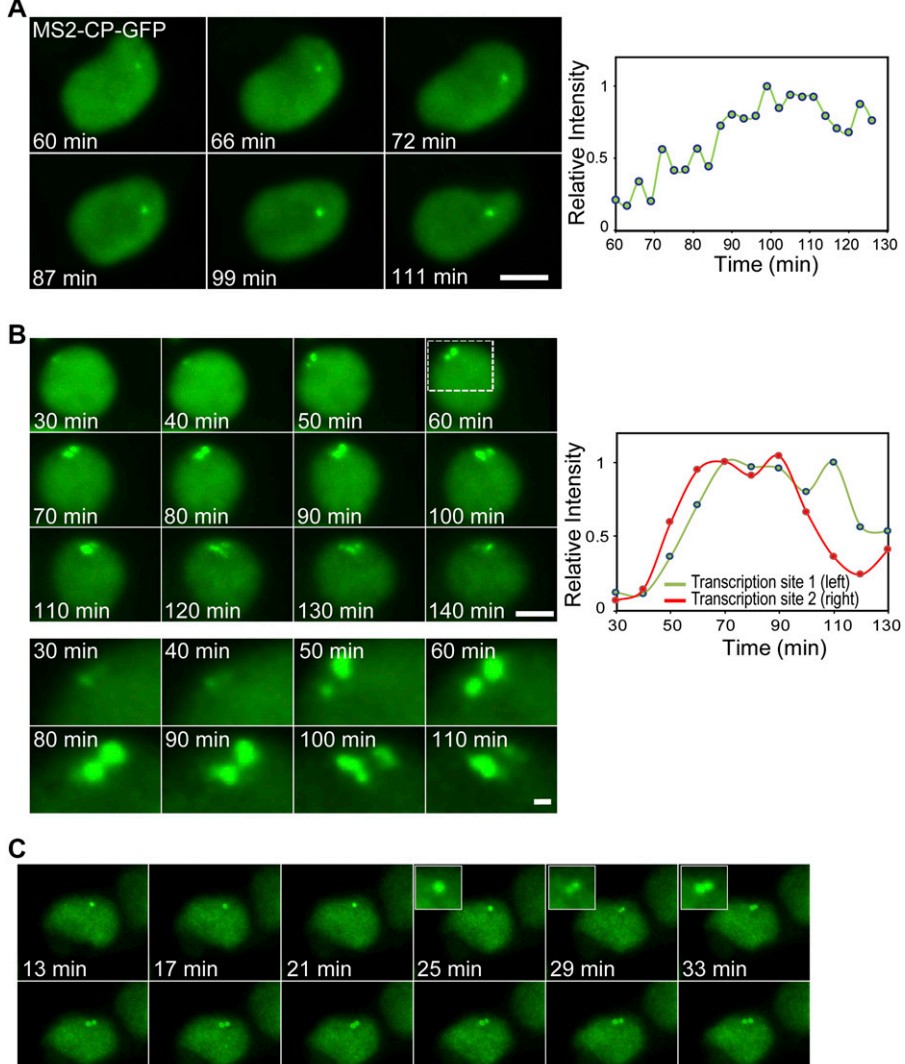

**Figure 5. Live-cell imaging of transcriptional activation following TSA treatment.**
**(A)** Frames from Video 1 showing the activation of the CCND1-MS2 allele under the control of the CMV promoter during TSA treatment. The cell is expressing MS2-CP-GFP. The plot shows the levels of intensity on the active gene over time. Time 0 is the time of TSA addition to the cells. **(B)** Frames from Video 2 showing the activation of duplicated CCND1-MS2 alleles (CMV driven) during TSA treatment. **(C)** Frames from Video 3 showing an active single CCND1-MS2 gene (CMV driven), its transition to a duplicated state, and the subsequent activation of the duplicated CCND1-MS2 alleles during TSA treatment. Scale bar, 5 μm.

the HDAC inhibition conditions allow for an unusually rapid burst of transcription even during G2.

Epigenetic regulation is not the whole story though. The comparison of the transcriptional activity of the CMV promoter and the endogenous *CCND1* promoter, both transcribing the same gene from the exact same genomic locus, exemplifies that epigenetic control is only one side of the coin. The *CCND1* promoter was influenced by the TSA treatment but to much lower levels compared with the CMV promoter, which was highly affected by TSA. The *CCND1* promoter sequence is highly complex, containing many types of transcription factor binding sites (Klein & Assoian, 2008), determining the restricted expression control of this gene, which is based on response to incoming signaling pathways. On the other hand, the CMV promoter has fairly simple regulatory sequences, and therefore, epigenetic control would have a major influence on its activity state. Hence, the effect of inhibiting deacetylation on the *CCND1* gene was relatively marginal, specifically since transcription regulation on this gene is tightly controlled through *cis*-acting elements and *trans*-acting factors.

Taken together, these measurements lead us to postulate that potentially a maximum output of transcriptional activity can be reached at any point of the cell cycle, pending that the epigenetic control has been removed. Yet, since the cell actively controls transcription under normal conditions, these potential levels are usually not obtainable, and so usually, actual transcription levels are significantly lower. This would then mean that the cell is fine tuned to spend energy on transcription and does not waste cellular resources on unnecessary transcription events.

## Materials and Methods

### Cell culture

The HEK293 Flp-in cells (Cat. No. R-750-07; Invitrogen) expressing the *cyclin D1* gene either under the CMV promoter or the *cyclin D1* promoter (obtained from R.G. Pestell, Thomas Jefferson University

[Albanese et al, 1995]) were previously described (Yunger et al, 2010). The genes contained the original pSL-MS2x24 sequence repeats, and the mRNAs were detected with the MS2-GFP coat protein (Bertrand et al, 1998; Fusco et al, 2003). The cells were maintained in DMEM containing 10% FBS (HyClone Laboratories). Cells are routinely tested for mycoplasma using the Hy-Mycoplasma PCR kit (Life Technologies). Transfections were performed by calcium phosphate precipitation (for transient MS2-CP-GFP expression). TSA (Sigma-Aldrich) was added for 7 h—fixed cells: 30 or 100 nM as marked, living cells 100 nM. EdU staining was performed with the Click-iT EdU Alexa Fluor 647 Imaging kit (Invitrogen).

### Fluorescence microscopy and live-cell imaging

Widefield fluorescence images were obtained using the CellR system based on an Olympus IX81 fully motorized inverted microscope (60× PlanApo objective, 1.42 numerical aperture, or 100× objective, 1.40 numerical aperture) fitted with an Orca-AG charge-coupled device camera (Hamamatsu) driven by CellR software. The microscope is equipped with a CFP/YFP dual-band filter and a DAPI/FITC/Tx-Red Triple Band filter set with single band excitation filters and ET GFP, ET Cy3, and ET Cy5 filter cubes (Chroma). For time-lapse imaging, cells were plated on glass-bottom tissue-culture plates with collagen coating (MatTek) in medium containing 10% FCS at 37°C. The microscope is equipped with an on-scope incubator, which includes temperature and $CO_2$ control (Life Imaging Services). For long-term imaging of transcription-site activation, several cell positions were chosen and recorded by a motorized stage (Scan IM, Märzhäuser).

In live-cell experiments, cells were typically imaged in four dimensions (3D over time). For presentation of the movies, the 4D image sequences were transformed into a time sequence by choosing the best focus (highest intensity) plane at each time point, using in-house–generated ImageJ scripts (US National Institutes of Health)—see detailed explanation on how to perform the tracking and generate the plots of transcription site intensity over time in the "SpotTracker plug-in for computing the transcription site trajectory" paragraph in *Nature Protocols* (Yunger et al, 2013). To improve quality, movies were deconvolved using Huygens Essential II with the time series option (Scientific Volume Imaging). Tracking was performed using the tracking module of Imaris (version 6.4; Bitplane) or the ImageJ Spot Tracker plugin. Correction of cell movement during tracking was performed using the "correct drift" option in ImageJ. Bleaching correction was applied to time-lapse images.

### Fluorescence in situ hybridization

A detailed protocol on how to perform the RNA FISH experiments and subsequent quantifications can be found in *Nature Protocols* (Yunger et al, 2013), including information about the sequence of the probe against the MS2 region, probe design, and the fluorescent labels. Calibrations of the probes for quantifying RNA in fixed cells can be found in the protocol in (Shav-Tal et al, 2010). In this study, we used 40 ng of the Cy3-MS2 DNA probe per coverslip, synthesized and fluorescently labeled by IBA. Briefly, for the quantification of

the number of mRNAs on the transcription sites or in the total cell, 3D stacks (0.2 $\mu$m steps, 76 planes) of the total volume of the cells were collected. The 3D stacks were deconvolved and the specific signals of mRNPs were identified (Imaris). mRNP identification was performed in comparison to deconvolved stacks from native HEK293 cells not containing the *D1-MS2* integration, which therefore served as background levels of nonspecific fluorescence. No mRNPs were identified in control cells. The sum of the intensity of each mRNA particle and transcription sites was measured in the same cells using Imaris. The single mRNP intensities were pooled and the frequent value was calculated. The sum of intensity at the transcription site was divided by the average intensity of a single mRNP. This ratio provided the number of mRNAs associated with the transcription unit from the point of the MS2 region and onwards.

### Immunofluorescence

After RNA FISH, cells were fixed for 20 min in 4% PFA (in PBS) and then permeabilized with 0.5% Triton X-100 for 3 min at room temperature. After blocking with 5% BSA (in PBS), cells were immunostained for 1 h with a primary antibody, and after subsequent washes, the cells were incubated for 1 h with a secondary antibody. Coverslips were stained with Hoechst and mounted in p-Phenylenediamine mounting medium (Cat. No. P6001; Sigma-Aldrich). Antibodies used were rabbit anti–CENP-F (Abcam) and Alexa-488 anti-rabbit secondary antibody (Invitrogen).

### Distance analysis

Cells in FISH experiments were counterstained with the Hoechst DNA stain. The whole volume of the nucleus was imaged, and this information underwent 3D isosurface rendering using Imaris to obtain the center of mass for the nucleus, which was presented in the x, y, and z coordinates. The x, y, and z coordinates were also determined for each active gene. The distances between the active genes and the nuclear center (in micrometer) were calculated using the equation:

$$d = \sqrt{(X_2 - X_1)^2} + \sqrt{(Y_2 - Y_1)^2} + \sqrt{(Z_2 - Z_1)^2}$$

Similarly, the distances between active genes in the same nucleus were calculated from x, y, and z coordinates obtained after performing "surface-object" fill-in in Imaris for each active gene.

### Real-time PCR (qRT–PCR)

Total RNA was extracted from cells using the AurumTM Total RNA mini kit (Bio-Rad Laboratories Inc.). After reverse transcription using the qScript cDNA Synthesis kit (Quanta Biosciences), cDNA was amplified using the following primer pairs:

HA sense: 5′-ACGATGTTCCAGATTACGCT; cyclin antisense: 5′-CAGGTCTCCTCCGCCTTC; tubulin sense: 5′-GCCTGGACCACAAGTTTGAC; antisense: 5′-TGAAATTCTGGGAGCATGAC; 18s sense: 5′-TGTGCC-GCTAGAGGTGAAATT antisense: 5′-TGGCAAATGCTTTCGCTTT. RT–PCR was performed using PerfeCTa SYBR Green FastMix, ROX (Quanta Bio Sciences) on a CFX-96 system (Bio-Rad). Analysis was performed

with the Bio-Rad CFX manager. Relative levels of mRNA expression were measured as the ratio of the comparative threshold cycle to internal controls (tubulin and 18S) RNA. Experiments were repeated at least three times.

### Statistical analysis

All FISH experiments were performed at least three times and on different days. The numbers of cells analyzed in each case (n) is noted in the actual plots themselves. A two-tailed $t$ test was performed to determine the significance of differences, and $P$'s are noted in the figure legends.

## Supplementary Information

## Acknowledgements

This work was supported by European Research Council (Y Shav-Tal) and the Israel Science Foundation No. 493/15 (Y Shav-Tal).

### Author Contributions

S Yunger: conceptualization, data curation, formal analysis, validation, investigation, visualization, methodology, and writing—original draft.
P Kafri: validation, investigation, visualization, methodology, and writing—original draft.
L Rosenfeld: software, investigation, and visualization.
E Greenberg: data curation and investigation.
N Kinor: data curation and investigation.
Y Garini: resources, software, and methodology.
Y Shav-Tal: conceptualization, supervision, funding acquisition, methodology, project administration, and writing—original draft, review, and editing.

### Conflict of Interest Statement

The authors declare that they have no conflict of interest.

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
