## [Reviewer comments · Life Science Alliance]

Life Science Alliance

S-phase transcriptional buffering quantified on two different promoters

Sharon Yunger, Pinhas Kafri, Liat Rosenfeld, Eliraz Greenberg, Noa Kinor, Yuval Garini, and Yaron Shav-Tal

Corresponding author(s): Yaron Shav-Tal, Bar-Ilan University

Review Timeline:

Submission Date:	2018-05-10
Editorial Decision:	2018-06-07
Revision Received:	2018-09-05
Editorial Decision:	2018-09-07
Revision Received:	2018-09-09
Accepted:	2018-09-10

Scientific Editor: Andrea Leibfried

Transaction Report:

June 7, 2018

Re: Life Science Alliance manuscript #LSA-2018-00086

Prof. Yaron Shav-Tal
Bar-Ilan University
Mina & Everard Goodman Faculty of Life Sciences
Institute of Nanotechnology
Ramat Gan 52900
Israel

Dear Dr. Shav-Tal,

Thank you for submitting your manuscript entitled "S-phase transcriptional buffering quantified on two different promoters" to Life Science Alliance. The manuscript was assessed by expert reviewers, whose comments are appended to this letter.

As you will see, the reviewers appreciate that your data confirm important previous findings in mammalian cells and they support publication of a revised version in Life Science Alliance. We would therefore like to invite you follow reviewer #2's suggestions on how to further strengthen your data and to submit a revised version of your manuscript. The points raised all seem straightforward to address, but please get in touch in case you would like to discuss individual points further.

-- High-resolution figure, supplementary figure and video files uploaded as individual files: See our detailed guidelines for preparing your production-ready images, <http://life-science-alliance.org/authorguide>

-- Summary blurb (enter in submission system): A short text summarizing in a single sentence the study (max. 200 characters including spaces). This text is used in conjunction with the titles of papers, hence should be informative and complementary to the title and running title. It should describe the context and significance of the findings for a general readership; it should be written in

the present tense and refer to the work in the third person. Author names should not be mentioned.

B. MANUSCRIPT ORGANIZATION AND FORMATTING:

Full guidelines are available on our Instructions for Authors page, <http://life-science-alliance.org/authorguide>

Thank you for this interesting contribution to Life Science Alliance. We are looking forward to receiving your revised manuscript.

Sincerely,

Andrea Leibfried, PhD
Executive Editor
Life Science Alliance
Meyerohofstr. 1
69117 Heidelberg, Germany
t +49 6221 8891 502
e a.leibfried@life-science-alliance.org
www.life-science-alliance.org

Reviewer #1 (Comments to the Authors (Required)):

In this MS, Shav-Tal and colleagues applied a system for in-vivo visualization of gene transcription to examine how cells adjust their gene expression following replication. Previous work by this group, as well as by other groups, have shown that cells attenuate transcription from each duplicated allele, thereby maintaining a constant ('homeostatic') level of gene expression despite the biased change in gene dosage caused by gene duplication. Previous studies in yeast also implicated a role

for H3K56ac in enabling this buffering.

The authors now establish that also in mammalian cells, expression buffering during S phase depends on histone acetylation. While they do not study a specific acetylation, they show convincingly that cells which are subject to the general TSA inhibitor, are not able to maintain buffering, but maintain high expression of both replicated allele. They further show that this inhibitor increases the overall level of transcription, suggesting that some limiting factor is released. The data is truly beautiful and convincing. The question examined is important, and the results interesting.

I strongly recommend publication

Reviewer #2 (Comments to the Authors (Required)):

Understanding the transcriptional activity correlated to cell cycle and DNA packing states is important to the epigenetic field. Single molecule microscopy allowing high resolution and real-time measurement of gene transcription at single cell and single chromosome level is unique to study complicated biological systems. The manuscript "S-phase transcriptional buffering quantified on two different promoters" submitted by Yunger et. al. is an interesting application of the papers "Single-allele analysis of transcription kinetics in living mammalian cells" published in Nature Methods in 2010 and "Quantifying the transcriptional output of single alleles in single living mammalian cells" published in Nature Protocols in 2013 from the same lab. In this work, the authors quantitatively determined the transcriptional rate in promoter-, cell cycle- and DNA packing-dependent manner. This paper is technically sound and suitable for publication in Life Science Alliance if the following issues are addressed.

1. The high concentrations of mRNAs in Figure 2A lead to bright nuclei and cytoplasm (instead of spotty-like patterns) in RNA FISH. Were these images used to count the number of cellular mRNAs (Figure 2C)? If so, how many cellular RNAs were determined in these cells?
2. Figure 3: In the absence of TSA, the transcription rate of CCND1-MS2 at single site state is faster than the transcription rate at doublet states (8 ± 2 vs. 4 ± 3 transcripts/gene). However, it does not contribute to the total amount of CCND1-MS2 RNAs. The authors should discuss potential reasons in the manuscript.
3. Figure 5: It is not clear how these relative intensity plots were generated.
4. The terms describing cell cycle stage is not precise. The authors defined G1 stage by single transcription site and "after S" stage by duplicated transcription sites. However, the single transcription site can be found in both G1 and early S cell cycle stages (G1/early S). It is unclear what the "after S stage" means.
5. The authors claimed increased transcription caused by TSA treatment is dose-dependent, about 3~5 fold increase in nascent mRNA on the duplicated genes at different TSA concentration (Figure S2B). A common and sensitive approach (such as qRT-PCR) should be performed to confirm the sensitivity of imaging-based quantification.
6. While it might be obvious to FISH experts how the RNA FISH probes are labeled, it is not trivial for general readers who have little or no experience on FISH techniques. How to generate labeled FISH probes or where to purchase labeled FISH probes is important information to reproduce the experiments. Also, they used FISH probes conjugated with five fluorophores (three internal and one at each end). Was any control experiment (e.g., single molecule/dye bleaching steps) performed to demonstrate the average number of dyes incorporated into individual probes? Such information is important for quantification purpose.

Minor points:

7. Material and Method section should be revised with more details: (1) the source of original cell line (2) Filters/dichroic used in the microscope system? (3) At what temperature the immunofluorescence experiment was carried out? (4) Buffers used to make 4% PFA and 5% BSA in immunofluorescence experiments (5) The brand/name of the mounting medium (6) the version of Imaris software? (7) Were the cells tested for mycoplasma contamination and how?

8. Although insertion of MS2 cassettes is a well-known method to label mRNAs in living cells, modifications have been made to make MS2 aptamer and the coat protein suitable for different purposes. The authors should indicate which MS2 cassette was used in this study by either citing literature or providing the sequence information.

9. The authors never mention what TSA stands for.

Points raised by reviewer 2:

1. The reviewer is correct. In the original figure we focused on showing the transcription sites and the differences in them between control and TSA-treated cells. We did not focus on the mRNA signals, and agree that the mRNAs should be clearly shown as well. We now provide images that reflect the mRNA signals too, in cells that were used in the actual quantifications. We also added the deconvolved images that are used in the process of quantification. The number of mRNAs is mentioned in the legend.

2. We have added text with an explanation regarding the total numbers of mRNAs in the cells, as follows:

“We note that although there is a decrease in the transcriptional output of the alleles after replication from 8 ± 2 nascent mRNAs for the single alleles in G1, to 4 ± 3 nascent mRNAs for doublets after replication (before TSA treatment), the cellular levels of the mRNAs remain generally the same, since each of the two alleles is simultaneously producing half of the mRNAs that a single allele does.”

3. We now refer to the protocol for generating the intensity plots where this is explained in detail, in the Methods section.

4. This is a valid point. In order to be precise we have modified the text in the appropriate places from “after S” to “after replication of the alleles”.

5. As suggested we have added a qRT-PCR experiment (performed at least 3 times) in Fig. S2 and included information in the Methods section.

6. The information about the specific probe that we use, its sequence, labeling, and how we use it in detail, appear in our two very detailed protocol papers. Indeed, this was not clear enough in the Methods section of the manuscript and so we have now elaborated on where to find all the exhaustive information necessary to perform these experiments, in a way that will be clear also to people who are less knowledgeable with the FISH technique, such that anyone can repeat such experiments.

7. In continuation to the above we have added the requested details: (1) Source of original cell line – Flp-In HEK293 cells (Invitrogen, cat. no. R-750-07). (2) We listed the filters that we use on our microscope system. (3) IF was performed at room temperature. (4) PFA and BSA are dissolved in PBS. (4) Imaris (Bitpalne, version 6.4). (5) Mounting medium - *p*-Phenylenediamine (Sigma-Aldrich, cat. no. P6001). (6) Cells are routinely tested for mycoplasma using the Hy-Mycoplasma PCR kit (Life Technologies).

8. We have added to the methods section that the plasmids used in this study (and generated for our initial study Yunger et al. Nat. Methods 2010) contained the original MS2 aptamers (pSL-MS2x24 and MS2-GFP) generated in (Bertrand et al., 1998; Fusco et al., 2003).

9. Embarrassing but correct, this important information was lacking. Trichostatin A now added to first mention.

September 7, 2018

RE: Life Science Alliance Manuscript #LSA-2018-00086R

Prof. Yaron Shav-Tal
Bar-Ilan University
Mina & Everard Goodman Faculty of Life Sciences
Institute of Nanotechnology
Ramat Gan 52900
Israel

Dear Dr. Shav-Tal

Thank you for submitting your revised manuscript entitled "S-phase transcriptional buffering quantified on two different promoters". We appreciate the introduced changes and we would be happy to publish your paper in Life Science Alliance pending final minor revisions as outlined below.

- please provide the main article as a .docx file
- please add the description for panel D to the legend of figure S2
- please mention the arrows of figure S6D in the legend

Full guidelines are available on our Instructions for Authors page, <http://life-science-alliance.org/authorguide>

****Reviews, decision letters, and point-by-point responses associated with peer-review at Life**

Science Alliance will be published online, alongside the manuscript. If you do want to opt out of this transparent process, please let us know immediately.**

Sincerely,

September 10, 2018

RE: Life Science Alliance Manuscript #LSA-2018-00086RR

Prof. Yaron Shav-Tal
Bar-Ilan University
Mina & Everard Goodman Faculty of Life Sciences
Institute of Nanotechnology
Ramat Gan 52900
Israel

Dear Dr. Shav-Tal,

Thank you for submitting your Research Article entitled "S-phase transcriptional buffering quantified on two different promoters". It is a pleasure to let you know that your manuscript is now accepted for publication in Life Science Alliance. Congratulations on this interesting work.

The final published version of your manuscript will be deposited by us to PubMed Central (PMC) as soon as we are allowed to do so, the application for PMC indexing has been filed. You may be eligible to also deposit your Life Science Alliance article in PMC or PMC Europe yourself, which will then allow others to find out about your work by Pubmed searches right away. Such author-initiated deposition is possible/mandated for work funded by eg NIH, HHMI, ERC, MRC, Cancer Research UK, Teletthon, EMBL.

Please also see:

<https://www.ncbi.nlm.nih.gov/pmc/about/authorms/>

<https://europepmc.org/Help#howsubsmanu>

*****IMPORTANT:** If you will be unreachable at any time, please provide us with the email address of an alternate author. Failure to respond to routine queries may lead to unavoidable delays in publication.*******

DISTRIBUTION OF MATERIALS:

Again, congratulations on a very nice paper. I hope you found the review process to be constructive and are pleased with how the manuscript was handled editorially. We look forward to future exciting submissions from your lab.

Sincerely,
